# High-Pressure Processing of Different Tissue Homogenates from Pigs Challenged with the African Swine Fever Virus

**DOI:** 10.3390/v16040638

**Published:** 2024-04-19

**Authors:** Stefano Petrini, Andrea Brutti, Cristina Casciari, Davide Calderone, Michela Pela, Monica Giammarioli, Cecilia Righi, Francesco Feliziani

**Affiliations:** 1National Reference Centre for Pestiviruses and Asfivirus, Istituto Zooprofilattico Sperimentale Umbria-Marche “Togo Rosati”, Via Gaetano Salvemini, 1, 06126 Perugia, Italy; c.casciari@izsum.it (C.C.); m.pela@izsum.it (M.P.); m.giammarioli@izsum.it (M.G.); c.righi@izsum.it (C.R.); f.feliziani@izsum.it (F.F.); 2SSICA Stazione Sperimentale per l’Industria delle Conserve Alimentari, Fondazione di Ricerca Parma, 43121 Parma, Italy; andrea.brutti@ssica.it; 3Associazione Industriali delle Carni e dei Salumi (ASSICA), 20089 Milan, Italy; calderone@assica.it

**Keywords:** ASFV, HPP, pigs, homogenates, sterilization techniques

## Abstract

African swine fever (ASF) is a disease that is a growing threat to the global swine industry. Regulations and restrictions are placed on swine movement to limit the spread of the virus. However, these are costly and time-consuming. Therefore, this study aimed to determine if high-pressure processing (HPP) sanitization techniques would be effective against the ASF virus. Here, it was hypothesized that HPP could inactivate or reduce ASF virus infectivity in tissue homogenates. To test this hypothesis, 30 aliquots of each homogenate (spleen, kidney, loin) were challenge-infected with the Turin/83 strain of ASF, at a 10 ^7.20^ median hemadsorption dose (HAD)_50_/mL. Subsequently, eight aliquots of each homogenate were treated with 600 millipascal (600 MPa) HPP for 3, 5, and 7 min. Six untreated aliquots were used as the controls. Virological results showed a reduction in the viral titer of more than 7-log. These results support the validity of the study hypothesis since HPP treatment was effective in inactivating ASFV in artificially prepared samples. Overall, this study suggests the need for further investigation of other ASFV-contaminated meat products.

## 1. Introduction

African swine fever (ASF) is a viral disease belonging to the family *Asfarviridae.* The virus contains linear dsDNA genomes that are about 170–194 kbp in length. The etiological agent is composed of an internal nucleoprotein core structure, 70–100 nm in diameter surrounded by an internal lipid layer, and an icosahedral capsid 170–190 nm in diameter surrounded by an external lipid-containing envelope. The external envelope has a diameter of 175–215 nm and is important for infection. Virions are inactivated within 30 min at 60 °C but survive for years at 20 °C or 4 °C. They are sensitive to ether, chloroform and deoxycholate. The infectivity is stable over a wide range of pH values. At pH 4 or pH 13, some infectious viruses may survive treatment. Some sanitizers (1% formaldehyde for 6 days, 2% NaOH for 1 day) destroy the infectious agent; para-phenylphenol sanitizers are very effective. The virus is sensitive to irradiation [1].

The virus poses a threat to the global swine industry [2,3] and affects both domestic and wild pigs [4], initially spreading in epidemic patterns (at least in previously unaffected territories) and later assuming the characteristics of an endemic infection.

The infection can be transmitted by direct contact, ingestion, or by ticks of the genus *Ornithodoros* between infected and healthy animals but more importantly, the indirect transmission, mainly mediated by the “human factor” [5,6].

The virus was evidenced from West Africa to the Iberian Peninsula, remaining endemic for over 30 years, and was finally eradicated from Europe, except for the island of Sardinia, Italy. In 2007, the virus spread out of Africa again through the Caucasus to Europe, and is currently causing outbreaks in the Russian Federation and several neighboring countries such as the Baltic Republics, Poland, Czech Republic and Romania [5,6,7]. Recently, ASF has appeared in mainland Italy, and currently, there are several outbreaks [8,9].

The European Food Safety Authority (EFSA) has recognized that a substantial portion of ASF outbreaks originates from contaminated kitchen waste. As a result, international regulation mandates have not only placed restrictions on the movement of pigs from infected areas, but also restrictions on all pork-derived products. The economic consequences of these measures are devastating, particularly because they need to be applied for extended periods owing to the difficulties in eradicating the disease [10].

To date, there has been a growing interest in sterilization and sanitization techniques for food products that can overcome restrictions and allow for the commercialization of these products.

High-pressure processing (HPP) is an already developed technology for reducing different bacterial or viral pathogens in food [11,12].

The first HPP-sanitized products were sold in Japan in 1990; these included jams, fruit jellies, sauces and fruit juices. Currently, various products processed under high pressure are commercially available in many countries. These products include fresh fish, bivalves, ready-to-eat meals, vegetable products, juices, beverages, salsa, jams and drinkable yoghurt. This technology is widely used in the States [13] and Japan [12].

Previous HPP studies have demonstrated that plant bacteria, yeasts and molds can be inactivated when exposed to pressures ranging from 200 to 700 MPa, thereby destroying the permeability of their cell membranes. Significant inactivation (>5 log reduction) has been reported for human norovirus (HuNoV), feline calicivirus (FCV), murine norovirus (MNV-1), hepatitis A virus (HAV), human Rotavirus (URV), Coxsackievirus A9 (CAV9), avian influenza (AIV), herpes simplex virus (HSV-1), human cytomegalovirus (HCMV), vesicular stomatitis virus (VSV), and avian metapneumovirus (AMPV) [12].

The main advantage of this treatment is that it operates at “cold” temperatures, allowing for processed foods to maintain their organoleptic characteristics and remain almost unchanged.

However, HPP technology has never been tested against the ASF virus. Therefore, in this study, we hypothesized that HPP could inactivate or reduce ASFV infectivity in the artificially contaminated tissue homogenates of pigs.

## 2. Materials and Methods

### 2.1. Sample Selection

Peripheral blood mononuclear cells (PBMCs) were collected from 3-month-old castrated crossbred pigs (Danish Landrace × Danish Duroc) devoid of ASF antibodies. PBMSs were extracted from venous blood samples. The samples were collected from the jugular vein using EDTA-containing sterile tubes (Terumo Vacuette^®^, Rome, Italy) and 21 G sterile needles (Terumo Needle^®^, Rome, Italy). The samples were transported to the laboratory at room temperature within 1 h of collecting before PBMC isolation.

### 2.2. Sample Preparation for HPP Treatment

Spleen, kidney, and muscle tissues (loin) were harvested from a 12-month-old castrated male pig (Danish Landrace × Danish Duroc) free of ASF from an Italian slaughterhouse in Perugia. Each tissue was homogenized (Braun Minipimern Gmbh, Waiblingen, Germany) in 300 mL of 2% phosphate-buffered saline (PBS; Carlo Erba Reagents, Milan, Italy) for 5 min at 22 °C. The entire homogenate was infected with the Torino/83 strain of ASF (batch 01/20) at a dose of 10 ^7.20^ median HAD_50_/mL and a ratio of 1:10. The virus was titred using the PBMCs. Subsequently, 10 mL of each homogenate was aliquoted into a double-embossed plastic food bag (Mantucci s.r.l., Perugia, Italy) and exposed to a pressure of −0.75 bar using a vacuum (Laica, Vicenza, Italy). A total of 30 aliquots (Table 1) were prepared for each sample and stored at −80 °C until their transfer to the “Stazione Sperimentale Industria delle Conserve Alimentari–Fondazione di Ricerca (SSICA)” in Parma, Italy to undergo HPP treatment. Before the samples were transferred to the SSICA, three aliquots from each sample (Table 1) were tested to determine the viral titer for ASF (see Section 2.4). The number of samples in each tissue was determined using the sampling procedure as in an experimental clinical study, which was used to compare the proportions in terms of superiority. A positive sample was considered a study event by identifying the following parameters for estimating the number of animals: 100% proportion of occurrence of the study event in the control group, 20% proportion of occurrence of the study event in the experimental group, an alpha error of 1%, and a study power of 90%.

### 2.3. HPP Treatment

After checking the viral titers of the aliquots, the remaining 27 samples were transferred from National Reference Centre for Pestiviruses and Asfivirus Laboratories (Perugia, Italy) to SSICA (Parma, Italy) for HPP treatment. After HPP treatment, all samples were transferred to National Reference Centre for Pestiviruses and Asfivirus for ASFV isolation. The samples were transferred under BSL3 biosafety conditions and kept in a freezer at −80 °C.

At the SSICA, the tissue homogenates were subjected to HPP treatment using a pilot plant with high-pressure equipment. In particular, hyperbaric treatment was conducted using an Avure QFP 35^®^ system produced by Avure SPA Vasteras (Avure Technologies JBT Group, Middletown, OH, USA). The system has 35 L of useful volume and can reach 600 MPa pressure. From an empty chamber, the maximum pressure is reached in about 2 min and is decompressed practically immediately. The system allows for treatments to be carried out in under adiabatic conditions at controlled temperatures between 4 and 90 °C. The initial temperature during the study was 10 °C, increasing by 3 °C per 100 MPa to reach a final temperature of 28 °C at 600 MPa. A pressure of 600 MPa was applied at three different times (3, 5 and 7 min) using eight aliquots of tissue homogenates for each exposure time. Three aliquots of the samples were not treated with HPP and were used as the negative controls (Table 2).

### 2.4. Virus Isolation and Quantification

Three aliquots of each homogenate before and after treatment with HPP were evaluated using viral isolation tests. The samples were thawed and centrifuged at 850× *g* for 10 min at 4 °C (Eppendorf ^®^ 5810R Centrifuge, Milan, Italy) and the supernatant (pure to a dilution of 10^−6^) was used for seeding PBMCs on a 96-well plastic plate (CytoOne ^®^ Plate, Starlab LTD, Blakelands, UK). The cell cultures were derived from 3-month-old castrated crossbred pigs (Danish Landrace × Danish Duroc) devoid of antibodies against ASFV. The cells were maintained in a minimum essential medium (MEM; Euroclone, Milan, Italy) and 30% of homologous pig serum at 37 °C with 5% CO_2_. After 4 days, the PBMCs were inoculated with each homogenate from pure to a dilution of 10^−6^. After 24 h, fresh homologous pig erythrocytes were added to a 1% buffer salt solution (Carlo Erba Reagents, Milan, Italy). The PBMCs infected with ASFV (Torino/83 strain of ASF batch 01/18) were used as the positive controls. The PBMCs free from ASFV were used as the negative controls. The plates were read daily for 7 days to check for the presence of viral hemadsorption. The titers of the positive homogenates were calculated using the Karber method [14]. Real-time PCR for ASFV [15] was performed on all homogenates to confirm the presence or absence of the virus in PBMCs.

## 3. Results

After packing the homogenates, the viral titer of the control ranged from 10 ^5.12^ to 10 ^5.53^ HAD_50_/mL. All samples that underwent HPP treatment, regardless of the time, showed negative results by virus isolation in PBMCs and confirmed by real-time PCR. Finally, the negative control, which did not undergo HPP treatment, showed titers ranging from 10 ^3.53^ to 10 ^5.53^ (Table 3).

## 4. Discussion

African swine fever is a disease that can be introduced into a country free of infection through the illegal import of contaminated pig products or by trading from ASF-infected areas [16].

Regarding virus resistance in Italian cured meats, Petrini et al. [17] demonstrated the effect of the dry curing process on the inactivation of the ASF virus in three different Italian dry-cured meat products prepared from experimentally infected pigs slaughtered at the peak of viremia. The results evidenced that the ASF virus was detected by in vivo experiments for up to 18, 60 and 83 days of curing in Italian salami, pork belly and loin, respectively [17].

In this context and as a result of the spread of the ASF in Europe, the European Union recently adopted Regulation (EU) 2023/594, establishing special control measures for African swine fever. In particular, the regulation applies to animal products, by-products of pig origin, and fresh meat and meat products, including casings, also from feral pigs [18].

As a result, alternative techniques to control the spread of the virus with products of pig origin have become a topic of interest to allow for the marketing of these products.

In a previous study [19], it was demonstrated that the cells of the A459 line infected with HEV and subjected to the pressure of 400 MPa applied for 1 min reduced the viral load of the virus by 2-log. Applying HPP to contaminated pâté protected it from HPP treatment [19]. The HPP inactivation of other foodborne viruses, including norovirus (HuNoV) [20] and hepatitis A (HAV), has been investigated [21]. Kingsley et al. report that HAV was stable in a cell culture at a pressure of 300 MPa, and inactivation increased by 6-log when the pressure increased from 300 to 450 MPa. Conversely, tests conducted on the feline calicivirus (FCV) showed a 3-log reduction when applying a pressure of 200 MPa [21]. In contrast, a high sensitivity to HPP was demonstrated in a study conducted on different HuNoV genotypes [20]. Indeed, using a pressure of 600 MPa at 4 °C for 5 min, two genotypes were inactivated, while one was reduced by 2.4-log. A similar trend to HuNoVs was observed for picornaviruses, as it was demonstrated that a pressure of 2.4 kbar for 1 h does not reduce neither poliovirus and rhinovirus, while the same pressure reduces the virus responsible for foot-and-mouth disease (FMD) by 4-log [22]. Finally, it has been demonstrated that 1–3 kbar pressures completely inactivate the human rhinovirus [23]. However, the effectiveness of HPP treatment depends on the nature of the virus, viral particle size, thermodynamic stability, viral receptors, structural proteins, amino acid composition and isoelectric point [20].

In this study, we hypothesized that different tissue homogenates experimentally infected with the strain Turin/83 of ASF at a dose of 10 ^7.20^ HAD_50_/mL would be inactivated by HPP treatment. For this purpose, we used different HPP inactivation times on homogenates from the spleen, kidney and muscle tissue (loin). The results of this study demonstrated that approximately a 7-log/mL reduction was achieved in the viral titer treated with ASF at 600 MPa with 3, 5 and 7 min hold times. These results agree with the previous paper regarding the efficacy of HPP against the reduction in the viral titer of different viral pathogens, as mentioned above.

Therefore, it can be deduced that the HPP treatment used in this study resulted in the inactivation of viral infectivity. This deduction was based on the viral isolation tests performed after HPP treatment which resulted in negative outcomes. The viral isolation was confirmed by real-time PCR. Consequently, it is speculated that the pressure exerted on the viral particles (600 MPa) resulted in their inactivation. Following this, it can be assumed that the inactivated viral particles (i) have changed size, (ii) condensed to nuclear material, (iii) altered protein envelope and (iv) broken nuclear walls [12,24].

This preliminary study demonstrated the efficacy of HPP treatment as a sanitizing method for products contaminated with ASF, even with short exposure times (3 min). This preliminary research encourages further investigation to verify the efficacy of HPP treatment on different matrices, particularly in processed cured meat products.

From this perspective, this technique could be employed to exempt trade restrictions imposed on areas affected by ASF infection. Given the vulnerability of these areas to the introduction of the ASF virus and the potential consequences of trade blockade resulting from restrictive measures, the adoption of HPP treatment could offer a promising solution.

It should be remembered that the HPP system is more effective as the water content of the treated material increases; the samples under study (organ homogenates) certainly contain much more water than cured products, so it will be important to check whether the good results obtained in this work can also be confirmed in cured products.

Future studies will be carried out on different pork products using pressures below 600 MPa and evaluating the efficacy of the HPP treatment.

## 5. Conclusions

In this study, we demonstrated the efficacy of HPP treatment in different homogenates previously infected with the strain Turin/83 of the ASF virus. Further studies will be conducted on other pork products infected with ASF.

## Figures and Tables

**Table 1 viruses-16-00638-t001:** Type of samples used in the experiments.

Pig Tissue Sample *	Sample Homogenates (mL) **	Virus Infection	Viral Dose (HAD_50_/mL)	Total Aliquots ***	Aliquots Used for Viral Titer Control ^
Spleen	300	Torino/83 strain of ASF	10 ^7.20^	30	3
Kidney	300	Torino/83 strain of ASF	10 ^7.20^	30	3
Muscle tissue (loin)	300	Torino/83 strain of ASF	10 ^7.20^	30	3

* Collected from a pig at a slaughterhouse in Perugia; ** each homogenized tissue was prepared in 2% PBS; HAD_50_: median hemadsorption dose/mL; *** each aliquot was prepared with 10 mL of the tissue homogenates; ^ before treatment with HPP.

**Table 2 viruses-16-00638-t002:** Samples used for ^1^HPP treatment with their exposure times.

Sample Homogenate	HPP Exposure Time (Min) *			Aliquots with No Treatment (Negative Controls)	Total Aliquots
	3	5	7		
Spleen	8 **	8	8	3	27
Kidney	8	8	8	3	27
Muscle tissue (loin)	8	8	8	3	27

* A pressure of 600 MPa was applied; ** number of aliquots tested; HPP: high-pressure processing.

**Table 3 viruses-16-00638-t003:** ASFV isolation from different homogenates infected with the Torino/83 strain of ASFV.

Homogenate	Aliquots Used for Viral Titer Control *	Aliquots Used for HPP Treatment **			Aliquots with No Treatment (Negative Control)
		3	5	7	
Spleen	5.12 ^	Neg.	Neg.	Neg.	3.53
Kidney	5.53	Neg.	Neg.	Neg.	5.53
Muscle tissue (loin)	5.20	Neg.	Neg.	Neg.	4.95

* No. three aliquots for each homogenate used before HPP treatment; ** No. eight aliquots for each exposure time; ^ reciprocal value of the negative log of the HAD_50_/mL (group mean value). Neg., negative result by virus isolation in PBMCs and confirm by real-time PCR.

## Data Availability

No new data were created or analyzed in this study. Data sharing is not applicable to this article.

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
