# Peer review of "High-Pressure Processing of Different Tissue Homogenates from Pigs Challenged with the African Swine Fever Virus"

_viruses, 2024, doi:10.3390/v16040638_

Round 1

Reviewer 1 Report

Comments and Suggestions for Authors

Some points to reconsider for further investigations:

·         is the method expensive?

·         detection of the virus in foods even though these viruses may have been inactivated

·         risk of incomplete inactivation

·         advantage, you also catch other viruses

·         interaction between food components such as fat, protein, salt and viral capsid

Author Response

Reviewer 1:

Some points to reconsider for further investigations:

R1. Is the method expensive?

A - Dear reviewer, thank you for your suggestions. The method applied in this study was not expensive. However, the instrumentation for HPP has a high cost. The HPP technology could be amortized by a large number of samples.

R1. Detection of the virus in foods even though these viruses may have been inactivated?

A - Dear reviewer, thank you for your suggestions. The aim of the study was precisely to test the possibility of inactivating the virus through the treatment. We did not carry out biomolecular tests just because it was expected that traces of the genome would be detected, whereas through the viral isolation test it was possible to verify the residual infection activity of the virus after treatment..

R1. Risk of incomplete inactivation

A - Dear reviewer, thank you for your suggestions. Under our experimental conditions, no viral replication activity could be detected in vitro

R1. Advantage, you also catch other viruses

A - Dear reviewer, thank you for your suggestions. It is not excluded that other viruses are inactivated in the HPP process. However, since the aim of this study was to evaluate the inactivation of the ASF virus, we cannot demonstrate the inactivation of other viruses. However, there are specific studies reported in the literature to verify the effect of treatment on other viruses

R1. Interaction between food components such as fat, protein, salt and viral capsid

A - Dear reviewer, thank you for your suggestions. The treatment is normally used to break down the bacterial load in food products. The influence of various factors such as % active water has been studied in this context. However, this aspect was not the focus of our study. Certainly further investigations need to be carried out to understand the best conditions for using the treatment..

Reviewer 2 Report

Comments and Suggestions for Authors

Indeed, the message of the manuscript is very provoking: "Italy processing pork from ASF infected pigs". The described technology proposes for food products inactivation, but not for destroying of infected carcasses during disease eradication. It means that authors suspect that pork products produced in Italy from infected pigs and situation with ASF is out of control. My advice - think twice before delivering this message to the public. 

"Further studies will be conducted on other pork products infected with ASF". I hope this will be not commercial products.

Otherwise, if products are safe and produced from ASF-free animals there no reason to apply this technology for virus inactivation.

From the methodology point of view, it will be extremely difficult to validate that technology because high pressure reduces the titer of the virus but not guarantee complete inactivation and if the titer will be higher or the amount of the products will be bigger it is not guarantee the virus inactivation.

"showed 158 negative results by virus isolation on PBMCs and confirmed by real-time PCR."

I did not get the point: samples became also PCR negartive after High Pressure treatment? It is important to include PCR results including Ct values before and after treatment.

Manuscript written in English and "PSA" need to be translated in English too.

"A spleen, kidney, and muscle tissue (loin) were harvested from a 12-month-old cas- 92 trated male pig (Danish Landrace × Danish Duroc) free of PSA from an Italian slaughter- 93 house in Perugia. Each tissue was homogenized"

The model is incorrect in my opinion. Authors simply can chaleng pigs with current ASF strain, circulation in Italy and collect the samples from infected pig.

In conclusion, even with HPP treatment it does not look good idea to use for marketing statment that products in Italy made from potencially sick pigs, even if ASFV had been inactivated.

Comments on the Quality of English Language

English need to be improved

Author Response

Reviewer 2

R2. Indeed, the message of the manuscript is very provoking: "Italy processing pork from ASF infected pigs". The described technology proposes for food products inactivation, but not for destroying of infected carcasses during disease eradication. It means that authors suspect that pork products produced in Italy from infected pigs and situation with ASF is out of control. My advice - think twice before delivering this message to the public. 

A - Dear reviewer, thank you for your suggestions. To date, Italy does not process pork from ASF-infected pigs. According to European legislation, outbreaks of ASF are destroyed and the meat is not used for the production of meat products. The objective of the study is to verify the possibility of offering additional health guarantees to the markets by offering sanitised products, but without calling into question the European Union's rules for fighting ASF. Therefore, in this study, we hypothesised that HPP could inactivate or reduce ASFV infectivity in artificially contaminated tissue homogenates of pigs. The results demonstrated this.

R2. "Further studies will be conducted on other pork products infected with ASF". I hope this will be not commercial products.

A - Dear reviewer, thank you for your suggestions. No studies will be conducted on commercial products infected with ASF. Only experimentally infected products will be used for HPP studies.

R2. Otherwise, if products are safe and produced from ASF-free animals there no reason to apply this technology for virus inactivation.

A - Dear reviewer, thank you for your suggestions. Your reasoning is agreeable, but markets and international agreements follow other logics. Japan has already requested Italy to apply HPP treatment on some products, and other international agreements are based on this argument..

R2. From the methodology point of view, it will be extremely difficult to validate that technology because high pressure reduces the titer of the virus but not guarantee complete inactivation and if the titer will be higher or the amount of the products will be bigger it is not guarantee the virus inactivation.

A - Dear reviewer, thank you for your suggestions. Based on our experimental conditions, we have demonstrated no residual replication activity of the virus. This is what we were able to verify under the experimental conditions we were allowed to apply. However, further studies can investigate this aspect that you rightly suggest.

R2. "showed 158 negative results by virus isolation on PBMCs and confirmed by real-time PCR." I did not get the point: samples became also PCR negative after High Pressure treatment? It is important to include PCR results including Ct values before and after treatment.

A - Dear reviewer, thank you for your suggestions. The real-time PCR was performed after only one pass on the cell cultures that turned negative. We used this test as confirmation of negativity. We did not include the Ct values as they were negative and were less than ≥ 40. We did not perform the RT-PCR test on the samples because this would most likely have been positive and would have given no information about the residual infectious capacity of the material. With the isolation test, on the other hand, we were able to verify the lowering of the starting titre to below the sensitivity threshold of the method, which in any case remains a gold standard

R2. Manuscript written in English and "PSA" need to be translated in English too.

A - Dear reviewer, thank you for your suggestions. We have translated PSA into ASF in English in different parts of the article.

R2. "A spleen, kidney, and muscle tissue (loin) were harvested from a 12-month-old cas- 92 trated male pig (Danish Landrace × Danish Duroc) free of PSA from an Italian slaughter- 93 house in Perugia. Each tissue was homogenized" The model is incorrect in my opinion. Authors simply can challenge pigs with the current ASF strain, circulation in Italy and collect the samples from infected pigs.

A - Dear reviewer, thank you for your suggestions. This scientific research was carried out when there were no outbreaks of African swine fever in Italy.

R2. In conclusion, even with HPP treatment it does not look good idea to use for marketing statment that products in Italy made from potencially sick pigs, even if ASFV had been inactivated.

A - Dear reviewer, thank you for your suggestions. We hypothesised that HPP could inactivate or reduce ASFV infectivity in artificially contaminated tissue homogenates of pigs. The results demonstrated this. We have not produced ASFV positive material from ASFV derived from pigs.

Reviewer 3 Report

Comments and Suggestions for Authors

It's an interesting paper.

In the affiliation 1 missing Petrini's email.

in the Abstract, at line 16 there two dot after ASF virus. At line 20 there is minutes abbreviation, but is the first, it is necessary to put full word and then the abbreviation.

For the Keywords you used "," but the instruction suggest ";" it is necessary to change.

Introduction: at line 33 it is necessary to insert the abbreviation of minutes.

At line 51 there are two dot after Romania.

In the Material and Methods it's necessray to insert the ethical commitee for the blood sample by pigs.

At line 93 it is necessary to put full word for PSA.

At line 95 and 150 missing the data for the manufacturing company for PBS and buffer salt solution. .

At line 148 what is 4d? 4 days? it's not clear. 

At lines154-155 it's not clear the PCR method used, it is necessary to descibed it.

At the Table 3 it's a font  discrepancy in the fourth column, the title is not in bold.

At line 167 missing the reference of Petrini.

At line 170 in vivo it should be written in italic.

At line 175 there is too psace between products, and by-products.

At line 180 you refer to a previous study, but the reference is missing.

In this paper you describe the efficincy o this methods to reduce viral load, but does the food maintain its organoleptic characteristics?

Author Response

Reviewer 3

R3. In the affiliation 1 missing Petrini's email.

A - Dear reviewer, thank you for your suggestions. In affiliation 1, Petrini's e-mail was added.

R3. In the Abstract, at line 16 there two dot after ASF virus. At line 20 there is minutes abbreviation, but is the first, it is necessary to put full word and then the abbreviation.

A - Dear reviewer, thank you for your suggestions. In line 16, there is only one dot after the ASF virus. In line 20, the term MPa has been extended to Millipascal.

R3. For the Keywords you used "," but the instruction suggest ";" it is necessary to change.

A - Dear reviewer, thank you for your suggestions. The suggested changes have been made.

R3. Introduction: at line 33 it is necessary to insert the abbreviation of minutes.

A - Dear reviewer, thank you for your suggestions. The abbreviation of minutes has been added.

R3. At line 51 there are two dot after Romania.

A - Dear reviewer, thank you for your suggestions. At line 51, the dot after 'Romania' has been deleted.

R3. In the Material and Methods it's necessary to insert the ethical committee for the blood sample by pigs.

A - Dear reviewer, thank you for your suggestions. The ethics committee was not involved in this study, as the blood was collected as part of common veterinary practices that do not require special care and/or suffering for the pig.

R3. At line 93 it is necessary to put full word for PSA.

A - Dear reviewer, thank you for your suggestions. The term PSA has been changed throughout the manuscript with the term ASF (abbreviation for African swine fever).

R3. At line 95 and 150 missing the data for the manufacturing company for PBS and buffer salt solution. 

A - Dear reviewer, thank you for your suggestions. The company data have been inserted.

R3. At line 148 what is 4d? 4 days? it's not clear. 

A - Dear reviewer, thank you for your suggestions. "4d" indicates 4 days. However, we have integrated 'd' in “days” in all necessary sections.

R3. At lines154-155 it's not clear the PCR method used, it is necessary to descibed it.

A - Dear reviewer, thank you for your suggestions. The method we applied in real-time PCR is the one described by King et al., 2003 (see below). It has been identified in bibliographic citation no. 15.

“King, D.P.; Reid, S.M.; Hutchings, G.H.; Grierson, S.S; Wilkinson, P.J.; Dixon, L.K.; Bastos, A.D.; Drew, T.W. 282. Development of a TaqMan PCR assay with internal amplification control for the detection of African swine fever virus. J. virol. Methods 2003, 107, 53-61”

R3. At the Table 3 it's a font  discrepancy in the fourth column, the title is not in bold.

A - Dear reviewer, thank you for your suggestions. The title in the fourth column has been changed to bold.

R3. At line 167 missing the reference of Petrini.

A - Dear reviewer, thank you for your suggestions. Reference No. 17 was added after Petrini.

R3. At line 170 in vivo it should be written in italic.

A - Dear reviewer, thank you for your suggestions. In line 170, the term in vivo was written in italics.

R3. At line 175 there is too space between products, and by-products.

A - Dear reviewer, thank you for your suggestions. On line 175, the space has been reduced to one character.

R3. At line 180 you refer to a previous study, but the reference is missing.

A - Dear reviewer, thank you for your suggestions. Reference No. 19 has been added to line 180.

R3. In this paper you describe the efficincy o this methods to reduce viral load, but does the food maintain its organoleptic characteristics?

A - Dear reviewer, thank you for your suggestions. The aim of this manuscript was not to evaluate the organoleptic characteristics of homogenates, but was to evaluate the HPP inactivate or reduce ASFV infectivity in artificially contaminated tissue homogenates of pigs. In any case, the treatment is widely used to reduce the bacterial load in food products and an extensive literature is available on the preservation of organoleptic qualities.

Round 2

Reviewer 2 Report

Comments and Suggestions for Authors

I do not have additional questions, it's good that manuscript will be published together with authors answers to reviewers comments because these answers more informative then manuscript by itself.